# The Effects of Using Wood Chips and Slash in Reducing Sheet Erosion on Forest Road Slopes

**Yilmaz Turk**

Department of Forest Engineering, Faculty of Forestry, Duzce University, Duzce 81620, Turkey;
yilmazturk@duzce.edu.tr; Tel.: +90-533-5474327

**Abstract:** This study compared the use of wood chips and slash to reduce the loss of sediment on newly constructed forest road slopes and investigated the annual amount of sediment loss on bare forest road slopes. A runoff block (sample field) was established for each of the four designated test sites (two cutslopes and two fillslopes). Each block had three runoff plots. One of the runoff plots was left empty for the control (CNT), while wood chips (C) and slash (S), respectively, were deposited in the other two. A total of 108 water samples were taken from the test sites and the amount of their suspended sediment calculated in the laboratory. As a result of this study, it was determined that the amount of soil loss in the control plots was about 1.26 times higher than in the slash plots and 2.21 times higher than in the wood chips plots. According to the results of variance analysis on the amounts of sediment, a statistically significant difference was found between the suspended sediment quantities transported on the road slopes ($p < 0.05$). However, no statistically significant difference between the suspended sediment quantities transported in the plots and the other variables of aspect, gradient or road slope was revealed by the $t$-test ($p > 0.05$).

**Keywords:** road slopes; sheet erosion; wood chips; slash; rehabilitate; Turkey

---

## 1. Introduction

Forest roads are the most important infrastructural facilities for the utilization of renewable natural forest resources. In order to achieve the goal of planning sustainable forestry activities, a road network must be established. Forest roads interact within many technical, economic, and environmental specifications in order to fulfill these tasks [1].

Although roads are the first step in the development of forestry resources, they are also infamous for bringing about certain problems. Forest road construction may cause environmental problems if not planned with consideration for required protective measures or not implemented in the field using suitable techniques. One of the major environmental problems resulting from improper forest road construction is the acceleration of soil erosion. In a forest ecosystem, the fertile topsoil is rich in organic matter which provides bulk density, permeability and erosion-resistance. During the road construction stage this topsoil is stripped off, altering the surface slope. This changes the structure of the soil as it is exposed and compacted for the construction area. Therefore, soil loss by erosion and surface runoff is accelerated in both cutslope and fillslope surfaces of forest roads. These effects emerge as the most important factors in determining soil loss by erosion [2–4].

Usually, the main reason for increased erosion after the construction of forest roads can be listed as the elimination, in part or as a whole, of the protective flora along the way [5–7]. In addition, with increased cutslope length there is also an increasing rate of soil loss [8–10]. Moreover, possible causes of the accelerated erosion, which may take place following road construction on forested lands, include destruction or impairment of the natural soil structure and fertility, increased slope gradients created by the construction of cutslopes and fillslopes, decreased infiltration rates on

parts of the road, interception of subsurface flow by the road cut gradient, decreased shear strength, increased shear stress, cutslopes and/or fillslopes, and concentration of generated and intercepted water [2,5].

More than 90% of sediment production originates from forest roads and the greater part of this occurs in excavations and gradient slopes [11,12]. In addition, road length and road width have significant effects on sediment generation [13].

To date, many studies have been conducted related to the environmental and ecological effects of forest roads and their surfaces [14–16]. The losses of sediment in natural and deteriorated areas have been compared. Soil loss and runoff rates on forest roads were investigated with simulation rainfall [6,7,17]. In addition, the average annual sediment generation from a forest road network was estimated by using a sediment prediction model and GIS techniques [18]. However, studies investigating the effects of the methods used to reduce the surface erosion on forest road slopes are very limited.

A stable running surface is durable and non-degrading and offers protection against erosion during changing weather conditions. In order to minimize surface erosion on forest road slopes, stabilization of the slopes should begin after road construction. The first winter following the construction of the slopes is a critical period for stabilization. During the period before vegetation development, many failures arise due to the slopes. The first failures occur after heavy rains. If they are remedied quickly and effective repair methods applied, more serious damage, such as the roads becoming unusable, can be prevented. A good maintenance program should be applied until the slopes are fully stabilized. Slope stabilization is expensive; however, when done properly, the costs are met by the stabilization fulfilling its functions [19].

In general, the methods applied against slope erosion can be divided into three types: structural, vegetative and biotechnical. Structural methods include stone arches, dry stone structures, gabions, log crib revetments and timber retaining walls. Among the vegetative methods are living fencing, mesh fencing, live bush bundles, branch covers, ready lawns, the planting of seedlings, seed spraying and planting. Biotechnical methods can include plant walls and wicker covers [2,9,19].

Several studies [20–24] used solid waste compost in road construction to reduce runoff and erosion. In addition, wood-based materials were used for erosion mitigation in some studies [25,26].

The first aim of this study was to compare the effects of using of wood chips and slash in reducing the loss of sediment on newly constructed forest road slopes (both cut and fill). Next, the study aimed to investigate the annual amount of sediment loss on bare forest road slopes and, finally, to compare the sediment losses of cutslopes and fillslopes.

## 2. Materials and Methods

### 2.1. Study Area

The study was carried out in the Aksu District (40°47′08″–40°52′02″ N, 31°16′45″–31°26′22″ E) of the Western Black Sea province of Duzce, Turkey. The altitude of the study area, which has a northern aspect, ranges between 355 and 1365 m a.s.l. The average annual rainfall is 816.7 mm and temperatures range from −0.4 to 28.5 °C, with an annual average during the summer of 24.4 °C. The forests of the study area are managed as high forests and have mixed stands. Tree species in the mixture include *Fagus orientalis* Lipsky, *Quercus* spp. and *Carpinus betulus* L. The study area covers 5966.5 ha of forest land. Four test sites (two cutslopes and two fillslopes) were determined on a newly constructed road (code No. 038) within the boundaries of the Aksu Forest Enterprise Directorate.

### 2.2. Forest Road

The forest road design was conducted in accordance with Communiqué No. 292 of the General Directorate of Forestry of Turkey. According to this communiqué, forest road construction is carried out by arranging the slope surfaces on both sides of the road platform and road by processing the natural ground with excavators or dozers. Dozers were traditionally used for road construction; however,

these days, in compliance with environment protection, excavators are used. Secondary forest roads use a 1:1 ratio in cutslopes (up to 10:1 on rocky ground), a 2:3 ratio in fillslopes and a 1:3 ratio in 1 m wide ditches. The surface of fillslopes formed by the excavated material is stabilized using a 2:3 ratio. The surface of cutslopes is formed by excavating the natural ground and the surface of the slope formed by the excavated material remains exposed to erosive effects.

The study area has a total of 161.5 km of forest roads. The actual road density value of the working area is 21.89 m/ha and the road distance is 588 m. The investigated road was constructed in 2014. Hydraulic excavators were used in the construction of the forest road in the study. The platform of the forest road in the study area was on average 5.26 m wide with an average road slope of 8%. It was a low-volume, secondary forest road (Figure 1).

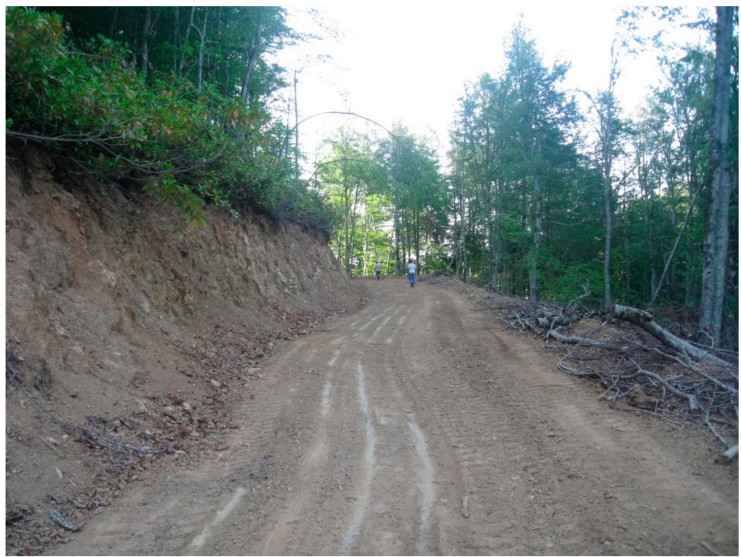

**Figure 1.** Forest road in study area.

## 2.3. Establishment of Runoff Plots

The field work began after the construction of the forest road. A runoff block (sample field) was established for each of the four designated test sites. Two of the blocks (fillslopes) had gradients of 67% and the other two (cutslopes) of 70%. Two of the blocks were planted facing north and the other two facing south (Table 1). Each block had three runoff plots. One of the runoff plots was left empty for the control (CNT), while wood chips (C) and slash (S), respectively, were deposited in the other two. The wood chips plots were laid with 12 kg (4 kg m$^{-2}$) of wood chips at a thickness of 1 cm. The wood chips were obtained from a private workshop in the study area region. In the slash plots, slash (branch/leaf) logging residue obtained from the test area was placed so as to cover the whole of the plot (1.5 kg m$^{-2}$). The 1 × 3 m plots were established with the long sides parallel to the direction of the slope. The sides and upper edges of the plots were constrained by 20 cm wide metal sheets, buried in the soil to a depth of 10 cm. The joints of the sheets were suitably connected and sealed with silicone to prevent the leakage of water at the sides. An inlaid structure was formed by inserting a plastic nylon layer at the bottom edge and the runoff water was deposited in a collection tank (Figure 2). The runoff water was measured by scale cylinders in the collection vessel and converted to mm units according to the plot area of 3 m$^2$ [27].

**Table 1.** General information related to the study area.

| Field Properties | Sample Fields (SF) | | | |
|---|---|---|---|---|
| | **SF 1** | **SF 2** | **SF 3** | **SF 4** |
| Aspect | Southeast | Southeast | Northwest | North |
| Gradient (%) | 70 | 67 | 70 | 67 |
| Altitude (m) | 831 | 829 | 744 | 725 |
| Road Slope | Cut | Fill | Cut | Fill |
| Soil Type | Sandy clayey mud | Sandy clayey mud | Sandy clayey mud | Sandy clayey mud |

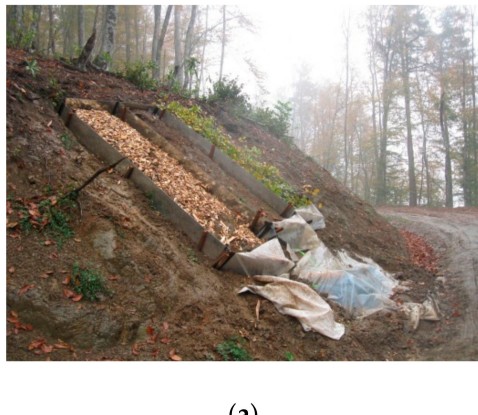
(**a**)

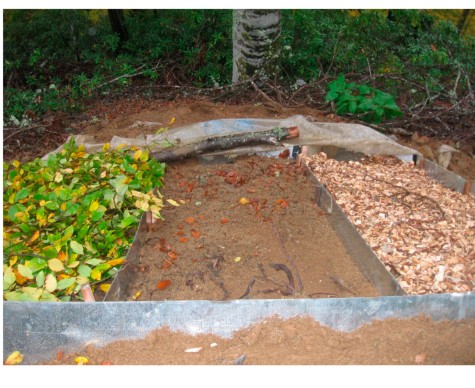
(**b**)

**Figure 2.** Establishment of runoff plots on road slopes: (**a**) Cutslopes (left—wood chips; center—control; right—slash); (**b**) fillslopes (left—slash; center—control; right—wood chips).

## 2.4. Water Sample Collection and Laboratory Analysis

The runoff passing through the soil after a rainfall, which accumulated in the collection tank, was measured using scale cylinders and the amount was recorded. When rainfall was not long-term or severe, monthly measurements were taken (excluding areas covered by snow). Nine measurements were taken from the study area throughout the period of October 2014–March 2016. A total of 108 water samples were collected from the test sites, 36 each from the control, wood chips and slash plots. After sediment that had settled at the bottom of the storage tanks was homogenized by shaking the storage tank, the measurement and recording process was completed. After completion, the storage tanks were completely drained.

In order to determine the amount of suspended sediment for each plot, a 0.5 L sample of water taken from the runoff storage tank was brought to the laboratory and put into the heating oven (for 1–2 days at 70 °C). After the runoff water evaporated, the samples of residual suspended sediment were weighed on an electronic precision balance. The specified value was calculated according to the total runoff amount accumulated in the storage tank and the total suspended sediment it carried (Figure 3). The materials coming from the slash plot collecting areas were trapped in a sieve in the collection tank and were not used in determining the amount of suspended sediment [28,29].

## 2.5. Statistical Analyses

The Kolmogorov-Smirnov test was used to determine whether or not the data were normally distributed. As the data did not show a normal distribution, normal distributions were provided by applying the square root transformation. Variance analysis was used to detect any differences in the amount of sediment and runoff among the plots in the experimental sites. The independent samples *t*-test was applied to find differences among the plots in the factors of aspect, gradient and road slope (cut and fill). All statistical analyses were performed using the SPSS statistical software (IBM SPSS Statistics for Windows, V. 22.0, IBM Corporation, Armonk, NY, USA).

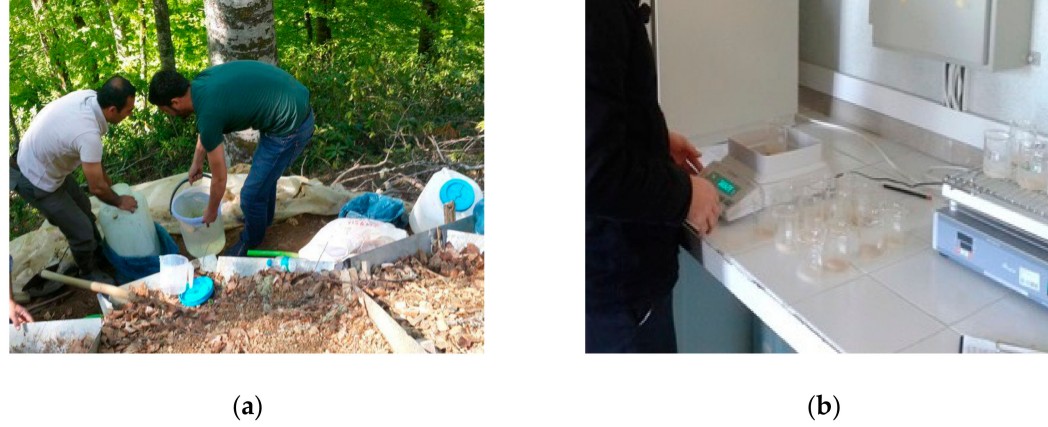

|       |       |
| :---: | :---: |
| (**a**) | (**b**) |

**Figure 3.** (**a**) Water sample collection; (**b**) water sample analysis at the laboratory.

## 3. Results and Discussion

### 3.1. Amount of Runoff and Suspended Sediment

Measurements during the period of October 2014–March 2016 were assessed by taking the nine plot measurements. The average runoff rates in the test site plots were 2.02 mm/m$^2$ in the control plots, 2.00 mm/m$^2$ in the slash plots and 1.94 mm/m$^2$ in the wood chip plots.

In the period covering runoff from the plots, the amounts of suspended sediment transported from the sample field plots are given in Table 2. The amount of sediment was found to be 6.42 g m$^{-2}$ in the control plots, 5.11 g m$^{-2}$ in the slash plots and 2.91 g m$^{-2}$ in the wood chips plots. These results indicated that wood chips application was better than slash for reducing the loss of sediment in the slopes. In their study, Foltz and Copeland [26] evaluated an erosion control product made by shredding on-site woody materials for mitigating erosion through a series of rainfall simulations. Results indicated that the wood product known as wood shreds reduced (ranging from 60 to nearly 100%) runoff and soil loss from both soil types (coarse and fine-grained), depending on the soil type, amount of concentrated flow and wood shred cover. In another study, Yanosek et al. [25] tested the effectiveness of two blends of manufactured wood strands of different lengths at controlling erosion with rainfall simulations in a laboratory. Wood strands were effective in delaying runoff, reducing runoff volume, and reducing sediment loss. In comparison to bare soil with no cover, sediment loss was reduced by at least 70% for all cover amounts tested and among each soil type, slope and flow event. This study and two other studies showed that the use of wood-based materials have mitigated erosion. One study indicated that a mulch as well as seed and fertilizer should be placed on roadside slopes if soil erosion is to be minimized during the first rainy season following road construction [12]. Another study pointed out that road segments where vegetation was cleared from the cutslope and ditch produced about seven times as much sediment as road segments where vegetation was retained [8].

An average of 35.29 g m$^{-2}$ year$^{-1}$ of soil loss was found in the control plots in this study. Studies performed on forest road cutslopes in different regions determined the amount of sediment loss in cutslopes to range from 0.5–37 kg m$^{-2}$ year$^{-1}$ [30–35]. The values obtained in the present study were lower than those in other studies. In some studies, when the amount of sediment loss in cutslopes and fillslopes was compared, more losses were found in the cutslopes [6,10,36]. However, one study reported that 60% of the sediments from roads are caused by fillslopes, 25% by the road surface and 15% by cutslopes [12]. In this study, the amount of sediment loss was found to be higher in fillslopes.

**Table 2.** Amount of suspended sediment in sample fields.

| | Amount of Suspended Sediment (g m$^{-2}$) | | | | | | | | | | | |
|---|---|---|---|---|---|---|---|---|---|---|---|---|
| | Sample Fields (SF) | | | | | | | | | | | |
| Sampling Periods | SF1 | | | SF2 | | | SF3 | | | SF4 | | |
| | C | CNT | S | C | CNT | S | C | CNT | S | C | CNT | S |
| October 2014 | 0.18 | 0.42 | 0.26 | 0.22 | 0.65 | 0.37 | 0.42 | 0.69 | 0.48 | 0.89 | 2.48 | 1.56 |
| November 2014 | 0.24 | 0.72 | 0.41 | 0.22 | 1.11 | 0.69 | 0.43 | 0.81 | 0.58 | 0.67 | 1.12 | 0.57 |
| December 2015 | 0.23 | 0.46 | 0.33 | 0.22 | 0.47 | 0.27 | 0.29 | 5.75 | 0.38 | 0.33 | 0.72 | 0.40 |
| May 2015 | 1.85 | 4.67 | 2.11 | 0.22 | 31.21 | 35.42 | 2.79 | 2.95 | 6.98 | 4.23 | 7.02 | 3.72 |
| June 2015 | 1.19 | 1.95 | 1.73 | 0.22 | 10.86 | 3.48 | 0.70 | 9.19 | 0.06 | 5.74 | 8.57 | 3.54 |
| July 2015 | 0.54 | 1.55 | 1.54 | 0.22 | 1.27 | 0.88 | 0.35 | 0.85 | 0.63 | 1.47 | 8.38 | 1.50 |
| October 2015 | 5.03 | 5.45 | 3.85 | 0.22 | 12.20 | 22.05 | 1.53 | 12.94 | 2.36 | 3.41 | 6.70 | 3.21 |
| November 2015 | 1.85 | 4.55 | 1.89 | 0.22 | 2.17 | 37.73 | 1.87 | 12.52 | 0.82 | 8.60 | 13.79 | 8.42 |
| March 2016 | 2.13 | 2.86 | 5.08 | 0.22 | 9.84 | 5.89 | 11.48 | 28.23 | 18.03 | 6.55 | 16.07 | 6.78 |
| Average | 1.47 | 2.52 | 1.91 | 4.40 | 7.75 | 11.86 | 2.21 | 8.21 | 3.37 | 3.54 | 7.20 | 3.30 |

C = Wood Chips Plot, CNT = Control Plot, S = Slash Plot.

*3.2. Plot Comparison Results*

According to the results of the 108 water samples taken to determine the soil loss from the plots in the test sites, it was determined that soil loss in the control plots was about 1.26 times higher than that in the slash plots and 2.21 times higher than that in the wood chips plots. In addition, it was determined that soil loss in the slash plots was about 1.76 times higher than that in the wood chips plots (Figure 4).

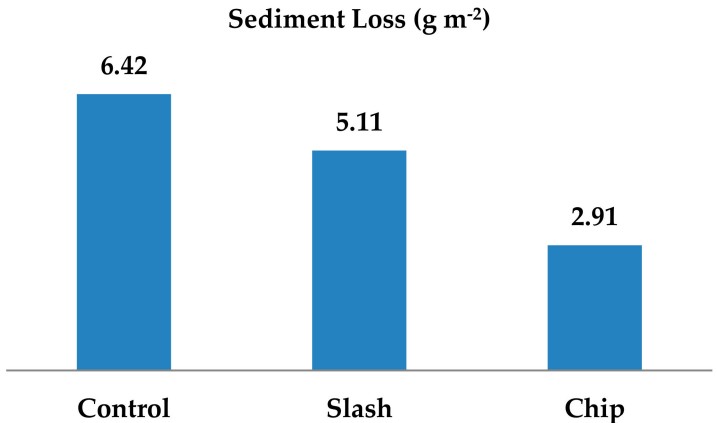

**Figure 4.** Soil loss in the test plots.

According to the results of variance analysis on the sediment amounts, a statistically significant difference was found between the suspended sediment quantities transported in the plots ($p < 0.05$). Suspended sediments transported in the control plots were the highest (1.409), while the lowest (1.136) were transported in the wood chips plots, with the transported slash plot amount falling between them (1.229). According to the results of variance analysis of the mean runoff rates, no statistically significant difference was found between the runoff quantities in the plots ($p > 0.05$) (Table 3). When the monthly amount of suspended sediment and monthly rainfall in plots are compared; statistically significant differences were found ($t = 6.096$, $p < 0.05$). This was due to the fact that the amount of suspended sediment was irregular. In this study, the plots were compared and the amount of runoff was found to be similar.

**Table 3.** Variance analysis results for transported suspended sediment.

| Parameters | Plots | Number of Samples | Average | Standard Error | *p* * |
|---|---|---|---|---|---|
| | CNT | 36 | 1.409 a [1] | 0.074 | |
| Suspended Sediment | C | 36 | 0.136 b | 0.062 | 0.029 |
| | S | 36 | 1.229 ab | 0.081 | |
| | CNT | 36 | 1.288 a | 0.102 | |
| Runoff | C | 36 | 1.274 a | 0.095 | 0.994 |
| | S | 36 | 1.286 a | 0.100 | |

* $p < 0.05$; [1]: The averages shown in the column with the same letter are statistically similar.

The differences between the suspended sediment quantities transported in the plots and the other variables were revealed by the *t*-test. There was no statistically significant difference among the suspended sediment quantities in the plots according to aspect (north or south), gradient or road slope (cut or fill) ($p > 0.05$) (Table 4).

**Table 4.** Results of *t*-test for other parameters in plots with transported suspended sediment.

| Parameters | Plots | Number of Samples | Difference in the Averages | Standard Error | F | t | *p* * |
|---|---|---|---|---|---|---|---|
| | CNT | 36 | −0.213 | 0.144 | 0.030 | −1.474 | 0.864 |
| Aspect | C | 36 | −0.043 | 0.125 | 0.003 | −0.345 | 0.955 |
| | S | 36 | 0.119 | 0.163 | 1.092 | 0.729 | 0.303 |
| | CNT | 36 | −0.140 | 0.147 | 0.570 | −0.952 | 0.456 |
| Gradient | C | 36 | −0.224 | 0.119 | 0.844 | −1.884 | 0.365 |
| | S | 36 | −0.285 | 0.157 | 1.497 | −1.812 | 0.230 |
| Road Slope | CNT | 36 | −0.140 | 0.147 | 0.570 | −0.952 | 0.456 |
| | C | 36 | −0.224 | 0.119 | 0.844 | −1.884 | 0.365 |
| | S | 36 | −0.285 | 0.157 | 1.497 | −1.812 | 0.230 |

* $p < 0.05$. F = Equality value of variances. t = Equality value of means.

## 4. Conclusions

In this study, sediment loss due to sheet erosion occurring on the forest road slopes was investigated and the use of wood chips and slash to reduce soil loss was compared. As a result of this study, it was determined that the amount of soil loss in the control plots was about 1.26 times higher than in the slash plots and 2.21 times higher than in the wood chips plots. In this study, in order to reduce the loss of sediment in slopes, wood chips application was shown to be better than slash. The amount of suspended sediment in the plots was found to be irregular. This may be due to different sediment fragmentation and dissolution in the plots during study.

According to the results of variance analysis on sediment amounts, a statistically significant difference was found between the suspended transported sediment quantities on the road slopes ($p < 0.05$). The transported suspended sediments were the highest (1.409) in the control plots, while the lowest (1.136) were transported in the wood chips plots. Sediment amounts transported in the slash plots fell between them (1.229). However, no statistically significant difference between the suspended sediment quantities transported in the plots and the other variables of aspect (north or south), gradient or road slope (cut or fill) was revealed by the *t*-test ($p > 0.05$).

With this study it has become clear that logging residues (wood chips and slash) can be used to reduce the sheet erosion that occurs on the road slopes after the construction of forest roads. Apart from the use of logging residues, other slope stabilization methods should be compared to determine the best method.

**Funding:** This research received no external funding.

**Acknowledgments:** This study was carried out with the support of Duzce University. The physical analyses were done in the laboratories of the Faculty of Forestry. We give special thanks to the Aksu Forest Enterprise Directorate for their permission and help. Special thanks are extended to Nuriye Peaci for English editing.

**Conflicts of Interest:** The authors declare no conflicts of interest. The founding sponsors had no role in the design of the study, the collection, analysis or interpretation of data, in the writing of the manuscript or in the decision to publish the results.

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
