# Peer review of "The Effects of Using Wood Chips and Slash in Reducing Sheet Erosion on Forest Road Slopes"

_forests, doi:10.3390/f9110712_

Round 1
Reviewer 1 Report
1. Update and upgrade Your Introduction part since more than 50% literatures in your reference list are more than 10 years old. Please study and take into consideration some of the manuscript listed below:
Ø Jones, O., Nyman, P., Sheridan, G., 2014. Modelling the effects of fire and rainfall regimes on extreme erosion events in forested landscapes. Stoch. Env. Res. Risk A. 1–11.
Ø Dalir, P., Naghdi, R., Gholami,V., 2014. Modeling of forest road sediment in the northern forest of Iran (Lomir Watershed). J Forest Sci. 60 (3), 109–114.
Ø Gholami, V., Khaleigh, M.R., 2013. The impact of vegetation on the bank erosion (case study: The Haraz River. Soil Water Res. 8 (4), 158–164.
Ø Huang, J., Zhao, X., Wu, P., 2013. Surface runoff volumes from vegetated slopes during simulated rainfall events. J. Soil Water Conserv. 38 (4), 283–295.
Ø Chamizo, S., Cantón, Y., Rodríguez-Caballero, E., Domingo, F., Escudero, A., 2012. Runoff at contrasting scales in a semiarid ecosystem: a complex balance between biological soil crust features and rainfall characteristics. J. Hydrol. 452, 130-138.
Ø De Oña, J., Osorio, F., Garcia, P.A., 2009. Assessing the effects of using compost–sludge mixtures to reduce erosion in road embankments. Journal of Hazard Mater. 164 (2-3), 1257-1265.
Ø Akay, A.E., Erdas, O., Reis, M., Yuksel, A., 2008. Estimating sediment yield from a forest road network by using a sediment prediction model and GIS techniques. Building Environ. 43 (5), 687-695.
Ø Jordan, A., Martinez-Zavala, L., 2008. Soil loss and runoff rates on unpaved forest roads in southern Spain after simulated rainfall, For Ecol Manag. 255 (3), 913-919.
2. You must take in to a consideration Type of soil also, since Foltz and Copeland 2009 and Yanosek et al. 2006 showed that it has important role in the soil erosion!!!
Ø Foltz, R.B., Copeland, N.S., 2009.Evaluating the efficacy of wood shreds for mitigating erosion. J. Eniron. Manag. 90 (2), 779–785.
Ø Yanosek, K.A., Foltz, R.B., Dooley, J.H., 2006. Performance assessment of wood strand erosion control materials among varying slopes, soil textures and cover amounts. J. Soil Water Conserv. 61 (12), 45–51.
3. Since You have collected data on the monthly basis please compare it with the monthly rainfalls and make a statistical analysis to see if there is any connections between them
4. You have written, at the page 2 line number 91, that the cutslopes ratio on the forest road is 1:1 but on the Fig. 1 this is not the case? Explain
5. When You are explaining the ditches, at the page 2 line number 91, the information’s regarding the shape and the depth are missing. Also I can not seen them on the Fig. 1. Explain
6. Instead of term range use distance page 3 line number 96
7. The sentence “As of 2018, all of the planned roads have 96 been constructed” does not have any sense. Please wrote the construction Year of the road where You have made investigation.
8. Instead of term slope use gradient page 3 line number 99
9. The term “without asphalt” does not have any sense. Please wrote the construction material of upper layer.
10. How You have defined the places for establishing the two runoff blocks?
11. Define the terms “Each rain” and “heavy rainfall” page 4 line number 131 and 132.
12. What happened with January, February, March, April 2015 and December, January, February 2016?
13. Define the term “precision scale” page 4 line number 141.
14. On the page 5, line number 170, how You can talk about this study and quote another author?
15. You are repeating the same data page 5, line number 174 and 175
16. It the table 2 correct CN to CNT
17. In conclusions part You must explain all try to explain all or most of illogical data from the table 2.
Author Response
Response to Reviewer 1 Comments
Point 1: Update and upgrade Your Introduction part since more than 50% literatures in your reference list are more than 10 years old. Please study and take into consideration some of the manuscript listed below:
Jones, O., Nyman, P., Sheridan, G., 2014. Modelling the effects of fire and rainfall regimes on extreme erosion events in forested landscapes. Stoch. Env. Res. Risk A. 1–11.
Dalir, P., Naghdi, R., Gholami,V., 2014. Modeling of forest road sediment in the northern forest of Iran (Lomir Watershed). J Forest Sci. 60 (3), 109–114.
Gholami, V., Khaleigh, M.R., 2013. The impact of vegetation on the bank erosion (case study: The Haraz River. Soil Water Res. 8 (4), 158–164.
Huang, J., Zhao, X., Wu, P., 2013. Surface runoff volumes from vegetated slopes during simulated rainfall events. J. Soil Water Conserv. 38 (4), 283–295.
Chamizo, S., Cantón, Y., Rodríguez-Caballero, E., Domingo, F., Escudero, A., 2012. Runoff at contrasting scales in a semiarid ecosystem: a complex balance between biological soil crust features and rainfall characteristics. J. Hydrol. 452, 130-138.
De Oña, J., Osorio, F., Garcia, P.A., 2009. Assessing the effects of using compost–sludge mixtures to reduce erosion in road embankments. Journal of Hazard Mater. 164 (2-3), 1257-1265.
Akay, A.E., Erdas, O., Reis, M., Yuksel, A., 2008. Estimating sediment yield from a forest road network by using a sediment prediction model and GIS techniques. Building Environ. 43 (5), 687-695.
Jordan, A., Martinez-Zavala, L., 2008. Soil loss and runoff rates on unpaved forest roads in southern Spain after simulated rainfall, For Ecol Manag. 255 (3), 913-919.
Response 1: Introduction part was updated and upgraded. Some of the suggested manuscripts have been added to the introduction part.
Point 2: You must take in to a consideration Type of soil also, since Foltz and Copeland 2009 and Yanosek et al. 2006 showed that it has important role in the soil erosion!!!
-Foltz, R.B., Copeland, N.S., 2009.Evaluating the efficacy of wood shreds for mitigating erosion. J. Eniron. Manag. 90 (2), 779–785.
-Yanosek, K.A., Foltz, R.B., Dooley, J.H., 2006. Performance assessment of wood strand erosion control materials among varying slopes, soil textures and cover amounts. J. Soil Water Conserv. 61 (12), 45–51.
Response 2: These two articles were included in the 3.1. Amount of runoff and suspended sediment Part and discussed. They are also included in the1. IntroductionPart.
Point 3: Since you have collected data on the monthly basis please compare it with the monthly rainfalls and make a statistical analysis to see if there is any connections between them
Response 3: Monthly data were collected from precipitation station closest to the study area. The runoff blocks were established in the same rainfall basin. The study plots are close enough together that precipitation amount, duration, and intensity can be considered the same. In addition, monthly amount of suspended sediment and monthly rainfall in plots are compared statistically. This was included in the 3.2. Plot comparison results Part. “When monthly amount of suspended sediment and monthly rainfall in plots are compared; statistically significant difference was found (t=6.096, P<0.05). This is due to the fact that the amount of suspended sediment was irregular. In this study, the plots were compared, the amount of runoff in the plots which are important in this study are similar. In this study, the amount of runoff was found to similar.”
Point 4: You have written, at the page 2 line number 91, that the cutslopes ratio on the forest road is 1:1 but on the Fig. 1 this is not the case? Explain
Response 4: This ratio can be up to 10:1 on rocky and stable grounds according to Communiqué no. 292 of the General Directorate of Forestry of Turkey. This ratio was included in the 2.2. Forest Road Part. Therefore, the slope ratio is high the Fig. 1.
Point 5: When You are explaining the ditches, at the page 2 line number 91, the information’s regarding the shape and the depth are missing. Also I can not seen them on the Fig. 1. Explain
Response 5: The information regarding of the ditches was corrected (a 1:3 ratio in 1 m-wide) according to Communiqué no. 292 of the General Directorate of Forestry of Turkey.
Point 6: Instead of term range use distance page 3 line number 96
Response 6: Instead of “range” term “distance” was used.
Point 7: The sentence “As of 2018, all of the planned roads have 96 been constructed” does not have any sense. Please wrote the construction Year of the road where You have made investigation.
Response 7: “As of 2018, all of the planned roads have been constructed” was deleted. Construction year of the investigated road was written. “The investigated road was constructed in 2014.”
Point 8: Instead of term slope use gradient page 3 line number 99
Response 8: Instead of “slope” term “gradient” was used.
Point 9: The term “without asphalt” does not have any sense. Please wrote the construction material of upper layer.
Response 9: “without asphalt.” was deleted. Since this study is related to the slopes, there is no need to give the upper layer properties (in my opinion).
Point 10: How You have defined the places for establishing the two runoff blocks?
Response 10: Two of the blocks were established on fillslopes and the other two were established on cutslopes. Two of the blocks were planted facing north and the other two facing south. The runoff blocks were established in the same rainfall basin.
Point 11: Define the terms “Each rain” and “heavy rainfall” page 4 line number 131 and 132.
Response 11: line number 131 and 132 was deleted (“Each rain” and “heavy rainfall”). Because this is the general method description in such studies. However, they were not mentioned in this study. Therefore, these statements are deleted from the article.
Point 12: What happened with January, February, March, April 2015 and December, January, February 2016?
Response 12: Since the study area was covered with snow and access to points was difficult in winter season. data could not be taken in these months. This was included in the article in the “2.4. Water sample collection and laboratory analysis part”.
Point 13: Define the term “precision scale” page 4 line number 141.
Response 13: Instead of “precision scale” term “electronic precision balance” was used. An electronic precision balance is device which defined as “a balance used to weigh quantities to a very precise number, usually up to one milligram”.
Point 14: On the page 5, line number 170, how You can talk about this study and quote another author?
Response 14: This was corrected. “However, it has been reported that 60% of the sediments from roads are caused by fillslopes, 25% by the road surface and 15% by cutslopes in their other study [14]. In this study, the amount of sediment loss was found to be higher in the fillslopes.” The citation was put in the wrong place.
Point 15: You are repeating the same data page 5, line number 174 and 175
Response 15: Repeating the same data were deleted.
Point 16: It the table 2 correct CN to CNT
Response 16: CN was corrected as CNT.
Point 17: . In conclusions part You must explain all try to explain all or most of illogical data from the table 2.
Response 17: These data were explained in conclusions part. “The amount of suspended sediment was found irregular in the plots. This may be due to different sediment fragmentation and dissolution in the plots during study.”
Point 18: Moderate English changes required.
Response 17: The English language has been revised.

Reviewer 2 Report
Overall, this is a well-written article of a small but useful study. There are, however, a few issues in the manuscript that should be addressed to make the study results more broadly applicable beyond the specific region in Turkey where the study took place. Detailed comments below:
Line Comment
56 “floor” – suggest “running surface”
76 The description of the study area needs to also include a description of the soil type/classification. This is key in understanding how the results of this study relate to studies conducted elsewhere.
91-92 Fillslope angles appear to be described as both 3:2 and 2:3.
128 It is unclear if only suspended sediment was collected in water samples or if sediment that had settled out at the bottom of the storage tanks was also collected, weighed, and recorded. Was there any settled sediment in storage tanks? If the study took place on a clay soil it could be reasonable to assume the majority of sediment might still be suspended in the water column. However, the ratio of suspended to settled sediment depends on time and it appears the amount of time between runoff event and water sample collection was inconsistent.
Additionally, was rainfall amount recorded at the study site? If so, it should be recorded here. If not, is there precipitation data on record for the region or somewhere nearby? At a minimum, precipitation amounts for each month should be given in table 2 alongside sediment amounts. Were the study plots close enough together that precipitation amount, duration, and intensity can be considered the same? Was all precipitation in the form of rain?
142 Are the “materials” from the slash plot dislodged slash that washed down to the collection tank? Was this only an issue with slash or chips as well?
159 “average runoff rates” assumes a time element (i.e. mm/m2/year)
161 It appears that “total sediment transported” was collected. This needs to be explained in the methods, as suspended sediment collection is the only method described.
173-174 These results were already presented above.
181 For SF2, “CNT” is shown as one of the sample fields vs. “CN” elsewhere. Is this a typo?
188 “…at a rather constant rate…” It appears from Table 2 that sedimentation amounts varied significantly by month.
Author Response
Response to Reviewer 2 Comments
Point 1: 56 “floor” – suggest “running surface”
Response 1: Instead of “floor” term “running surface” was used.
Point 2: 76 The description of the study area needs to also include a description of the soil type/classification. This is key in understanding how the results of this study relate to studies conducted elsewhere.
Response 2: Soil types of sample fields were included in the Table 1. General information related to the study area.
Point 3: Fillslope angles appear to be described as both 3:2 and 2:3.
Response 3: Fillslope ratio was fixed to 2:3.
Point 4: 128 It is unclear if only suspended sediment was collected in water samples or if sediment that had settled out at the bottom of the storage tanks was also collected, weighed, and recorded. Was there any settled sediment in storage tanks? If the study took place on a clay soil it could be reasonable to assume the majority of sediment might still be suspended in the water column. However, the ratio of suspended to settled sediment depends on time and it appears the amount of time between runoff event and water sample collection was inconsistent.
Additionally, was rainfall amount recorded at the study site? If so, it should be recorded here. If not, is there precipitation data on record for the region or somewhere nearby? At a minimum, precipitation amounts for each month should be given in table 2 alongside sediment amounts. Were the study plots close enough together that precipitation amount, duration, and intensity can be considered the same? Was all precipitation in the form of rain?
Response 4: These were explained in the 2.4. Water sample collection and laboratory analysis Part. “After sediment that had settled out at the bottom of the storage tanks was homogenized by shaking the storage tank, the measurement and recording process was completed. After all, the storage tanks were completely drained.”
Monthly data were collected from precipitation station closest to the study area. The runoff blocks were established in the same rainfall basin. The study plots are close enough together that precipitation amount, duration, and intensity can be considered the same. In addition, Monthly amount of suspended sediment and monthly rainfall in plots are compared statistically. This was included in the 3.2. Plot comparison results Part. “When monthly amount of suspended sediment and monthly rainfall in plots are compared; statistically significant difference was found (t=6.096, P<0.05). This is due to the fact that the amount of suspended sediment was irregular. In this study, the plots were compared, the amount of runoff in the plots which are important in this study are similar. In this study, the amount of runoff was found to similar.”
Point 5: 142 Are the “materials” from the slash plot dislodged slash that washed down to the collection tank? Was this only an issue with slash or chips as well?
Response 5: Only a small amount of wood chips was kept in the filter of collection tank. Wood chips were effective in delaying runoff.
Point 6: 159 “average runoff rates” assumes a time element (i.e. mm/m2/year)
Response 6: The averages in the measured period are given, so the year has not been added to the unit.
Point 7: 161 It appears that “total sediment transported” was collected. This needs to be explained in the methods, as suspended sediment collection is the only method described.
Response 7: “the total sediment transported” was deleted in the 3.1. Amount of runoff and suspended sediment Part. Because it was written by mistake.
Point 8: 173-174 These results were already presented above.
Response 8: Repeating the same data were deleted.
Point 9: 181 For SF2, “CNT” is shown as one of the sample fields vs. “CN” elsewhere. Is this a typo?
Response 9: CN was corrected as CNT.
Point 10: 188 “…at a rather constant rate…” It appears from Table 2 that sedimentation amounts varied significantly by month.
Response 10: "a rather constant rate" was cited from another study. This has been deleted.
Point 11: English language and style are fine/minor spell check required
Response 11: The English language has been revised.

Round 2
Reviewer 1 Report
Don't have any additional comments